# A LARGE-SCALE STUDY ON TRAINING SAMPLE MEMORIZATION IN GENERATIVE MODELING

## ABSTRACT

Many recent developments on generative models for natural images have relied on heuristically-motivated metrics that can be easily gamed by memorizing a small sample from the true distribution or training a model directly to improve the metric. In this work, we critically evaluate the gameability of such metrics by running a competition that ultimately resulted in participants attempting to cheat. Our competition received over 11000 submitted models and allowed us to investigate both intentional and unintentional memorization. To stop intentional memorization, we propose the "Memorization-Informed Fréchet Inception Distance" (MiFID) as a new memorization-aware metric and design benchmark procedures to ensure that winning submissions made genuine improvements in perceptual quality. Furthermore, we manually inspect the code for the 1000 top-performing models to understand and label different forms of memorization. The inspection reveals that unintentional memorization is a serious and common issue in popular generative models. The generated images and our memorization labels of those models as well as code to compute MiFID are released to facilitate future studies on benchmarking generative models.

## 1 INTRODUCTION

Recent work on generative models for natural images has produced huge improvements in image quality, with some models producing samples that can be indistinguishable from real images (Karras et al., 2017; 2019a;b; Brock et al., 2018; Kingma & Dhariwal, 2018; Maaløe et al., 2019; Menick & Kalchbrenner, 2018; Razavi et al., 2019). Improved sample quality is important for tasks like super-resolution (Ledig et al., 2017) and inpainting (Yu et al., 2019), as well as creative applications (Park et al., 2019; Isola et al., 2017; Zhu et al., 2017a;b). These developments have also led to useful algorithmic advances on other downstream tasks such as semi-supervised learning (Kingma et al., 2014; Odena, 2016; Salimans et al., 2016; Izmailov et al., 2019) or representation learning (Dumoulin et al., 2016; Donahue et al., 2016; Donahue & Simonyan, 2019).

Modern generative models utilize a variety of underlying frameworks, including autoregressive models (Oord et al., 2016), Generative Adversarial Networks (GANs; Goodfellow et al., 2014), flow-based models (Dinh et al., 2014; Rezende & Mohamed, 2015), and Variational Autoencoders (VAEs; Kingma & Welling, 2013; Rezende et al., 2014). This diversity of approaches, combined with the philosophical nature of evaluating generative performance, has prompted the development of heuristically-motivated metrics designed to measure the perceptual quality of generated samples such as the Inception Score (IS; Salimans et al., 2016) or the Fréchet Inception Distance (FID; Heusel et al., 2017). These metrics are used in a benchmarking procedure where "state-of-the-art" results are claimed based on a better score on standard datasets.

Indeed, much recent progress in the field of machine learning as a whole has relied on useful benchmarks on which researchers can compare results. Specifically, improvements on the benchmark metric should reflect improvements towards a useful and nontrivial goal. Evaluation of the metric should be a straightforward and well-defined procedure so that results can be reliably compared. For example, the ImageNet Large-Scale Visual Recognition Challenge (Deng et al., 2009; Russakovsky et al., 2015) has a useful goal (classify objects in natural images) and a well-defined evaluation procedure (top-1 and top-5 accuracy of the model's predictions). Sure enough, the ImageNet

benchmark has facilitated the development of dramatically better image classification models which have proven to be extremely impactful across a wide variety of applications.

Unfortunately, some of the commonly-used benchmark metrics for generative models of natural images do not satisfy the aforementioned properties. For instance, although the IS is demonstrated to correlate well with human perceived image quality (Salimans et al., 2016), Barratt & Sharma (2018) points out several flaws of the IS when used as a single metric for evaluating generative modeling performance, including its sensitivity to pretrained model weights which undermines generalization capability. Seperately, directly optimizing a model to improve the IS can result in extremely *unrealistic*-looking images (Barratt & Sharma, 2018) despite resulting in a better score. It is also well-known that if a generative model memorizes images from the training set (i.e. producing *non-novel* images), it will achieve a good IS (Gulrajani et al., 2018). On the other hand, the FID is widely accepted as an improvement over IS due to its better consistency under perturbation (Heusel et al., 2017). However, there is no clear evidence of the FID resolving any of the flaws of the IS. A large-scale empirical study is necessary to provide robust support for understanding quantitatively how flawed the FID is.

Motivated by these issues, we want to benchmark generative models in the "real world", i.e. outside of the research community by holding a public machine learning competition. To the extent of our knowledge, no large-scale generative modeling competitions have ever been held, possibly due to the immense difficulty of identifying training sample memorization in a efficient and scalable manner. We designed a more rigorous procedure for evaluating competition submissions, including a memorization-aware variant of FID for autonomously detecting cheating via intentional memorization. We also manually inspected the code for the top 1000 submissions to reveal different forms of intentional or unintentional cheating, to ensure that the winning submissions reflect meaningful improvements, and to confirm efficacy of our proposed metric. We hope that the success of the first-ever generative modeling competition can serve as future reference and stimulate more research in developing better generative modeling benchmarks.

Our main goal in this paper is to conduct an empirical study on issues of relying on the FID as a benchmark metric to guide the progression of generative modeling. In Section 2, we briefly review the metrics and challenges of evaluating generative models. In Section 3, we explain in detail the competition design choices and propose a novel benchmarking metric, the Memorization-Informed Fréchet Inception Distance (MiFID). We show that MiFID enables fast profiling of participants that intentionally memorize the training dataset. In Section 4, we introduce a dataset released along with this paper that includes over one hundred million generated images and manual labels obtained by painstaking code review. In Section 5, we connect phenomena observed in large-scale benchmarking of generative models in the real world back to the research community and point out crucial but neglected flaws in the FID.

## 2 BACKGROUND

In generative modeling, our goal is to produce a model $p_\theta(x)$ (parameterized by $\theta$) of some true distribution $p(x)$. We are not given direct access to $p(x)$; instead, we are provided only with samples drawn from it $x \sim p(x)$. In this paper, we will assume that samples $x$ from $p(x)$ are 64-by-64 pixel natural images, i.e. $x \in \mathbb{R}^{64 \times 64 \times 3}$. A common approach is to optimize $\theta$ so that $p_\theta(x)$ assigns high likelihood to samples from $p(x)$. This provides a natural evaluation procedure which measures the likelihood assigned by $p_\theta(x)$ to samples from $p(x)$ that were held out during the optimization of $\theta$. However, not all models facilitate exact computation of likelihoods. Notably, Generative Adversarial Networks (GANs) (Goodfellow et al., 2014) learn an "implicit" model of $p(x)$ from which we can draw samples but that does not provide an exact (or even an estimate) of the likelihood for a given sample. The GAN framework has proven particularly successful at learning models which can generate extremely realistic and high-resolution images, which leads to a natural question: How should we evaluate the quality of a generative model if we can't compute the likelihood assigned to held-out samples?

This question has led to the development of many alternative ways to evaluate generative models (Borji, 2019). A historically popular metric, proposed in (Salimans et al., 2016), is the Inception Score (IS) which computes

$$\mathrm{IS}(p_\theta) = \mathbb{E}_{x \sim p_\theta(x)}[D_{\mathrm{KL}}(\mathrm{IN}(y|x) \| \mathrm{IN}(y))]$$

where $\text{IN}(y|x)$ is the conditional probability of a class label $y$ assigned to a datapoint $x$ by a pre-trained Inception Network (Szegedy et al., 2015). More recently, (Heusel et al., 2017) proposed the Fréchet Inception Distance (FID) which better correlates with perceptual quality. The FID uses the estimated mean and covariance of the Inception Network feature space distribution to calculate the distance between the real and fake distributions up to second order. The FID between the real images $r$ and generated images $g$ is computed as:

$$\text{FID}(r, g) = \|\mu_r - \mu_g\|_2^2 + \text{Tr}\left(\Sigma_r + \Sigma_g - 2\left(\Sigma_r \Sigma_r\right)^{\frac{1}{2}}\right)$$

where $\mu_r$ and $\mu_g$ are the mean of the real and generated images in latent space, and $\Sigma_r$ and $\Sigma_g$ are the covariance matrices for the real and generated feature vectors. A drawback of both the IS and FID is that they assign a very good score to a model which simply memorizes a small and finite sample from $p(x)$ (Gulrajani et al., 2018), an issue we address in section 3.1.

## 3 GENERATIVE MODELING COMPETITION DESIGN

We designed the first generative model competition where participants were invited to generate realistic dog images given 20,579 images of dogs from ImageNet (Russakovsky et al., 2015). Participants were required to implement their generative model in a constrained computation environment to prevent them from obtaining unfair advantages. The computation environment was designed with:

- Limited computation resource (9 hours on a NVIDIA P100 GPU for each submission) since generative model performance is known to be highly related to the amount of computational resources used (Brock et al., 2018)
- Isolated containerization to avoid continuous training by reloading model checkpoints from previous sessions
- No access to external resources (i.e. the internet) to avoid usage of pre-trained models or additional data

Each submission is required to provide 10,000 generated images of dimension $64 \times 64 \times 3$ and receives a public score in return. Participants are allowed to submit any number of submissions during the two-month competition. Before the end of the competition, each team is required to choose two submissions, and the final ranking is determined by the better private score (described below) out of the two selected submissions.

In the following sections, we discuss how the final decisions were made regarding pretrained model selection (for FID feature projection) and how we enforced penalties to ensure the fairness of the competition.

### 3.1 MEMORIZATION-INFORMED FRÉCHET INCEPTION DISTANCE (MIFID)

The most crucial part of the competition is the performance evaluation metric to score the submissions. To assess the quality of generated images, we adopted the Fréchet Inception Distance (Heusel et al., 2017) which is a widely used metric for benchmarking generative tasks. Compared to the Inception Score (Salimans et al., 2016), the FID has the benefits of better robustness against noise and distortion and more efficient computation (Borji, 2019).

For a generative modeling competition, a good metric not only needs to reflect the quality of generated samples but must also allow easy identification of cheating with as little manual intervention as possible. Many forms of cheating were prevented by setting up the aforementioned computation environment, but even with these safeguards it would be possible to "game" the FID score. Specifically, we predicted that memorization of training data would be a major issue, since current generative model evaluation metrics such as IS or FID are prone to reward high scores to memorized instances (Gulrajani et al., 2018). This motivated the addition of a "memorization-aware" metric that penalizes models producing images too similar to the training set.

Combining memorization-aware and generation quality components, we introduced the Memorization-Informed Fréchet Inception Distance (MiFID) as the metric used for the competition:

$$\text{MiFID}(S_g, S_t) = m_\tau(S_g, S_t) \cdot \text{FID}(S_g, S_t)$$

where $S_g$ is the generated set, $S_t$ is the original training set, FID is the Fréchet Inception Distance, and $m_\tau$ is the memorization penalty which we discuss in the following section.

### 3.1.1 MEMORIZATION PENALTY

To capture the similarity between two sets of data – in our case, generated images and original training images – we started by measuring similarity between individual images. Cosine similarity, the inner product of two vectors, is a commonly used similarity measure. It is easy to implement with high computational efficiency (with existing optimized BLAS libraries) which is ideal when running a competition with hundreds of submissions each day. The value is also bounded, making it possible to intuitively understand and compare the degree of similarity.

We define the memorization distance $s$ of a target projected generated set $S_g \subseteq \mathbb{R}^d$ with respect to a reference projected training set $S_t \subseteq \mathbb{R}^d$ as 1 subtracted by the mean of minimum (signed cosine) similarity of all elements $S_g$ and $S_t$. Intuitively, lower memorization distance is associated with more severe training sample memorization. Note that the distance is asymmetric i.e. $s(S_g, S_t) \neq s(S_t, S_g)$, but this is irrelevant for our use-case.

$$s(S_g, S_t) := 1 - \frac{1}{|S_g|} \sum_{x_g \in S_g} \min_{x_t \in S_t} \frac{|\langle x_g, x_t \rangle|}{|x_g| \cdot |x_t|}$$

We hypothesize that cheating submissions with intentional memorization would generate images with significantly lower memorization distance. To leverage this idea, only submissions with distance lower than a specific threshold $\tau$ are penalized. Thus, the memorization penalty $m_\tau$ is defined as

$$m_\tau(S_g, S_t) = \begin{cases} \frac{1}{s(S_g, S_t) + \epsilon} & (\epsilon \ll 1), \quad \text{if } s(S_g, S_t) < \tau \\ 1, & \text{otherwise} \end{cases}$$

More memorization (subceeding the predefined threshold $\tau$) will result in higher penalization. Dealing with false positives and negatives under this penalty scheme is further discussed in Section 3.2.

### 3.1.2 PREVENTING OVERFITTING

In order to prevent participants of the competition from overfitting to the public leaderboard, we used different data for calculating the public and private score and we generalized the FID to use any visually-relevant latent space for feature projection. Specifically, we selected different pre-trained ImageNet classification models for public and private score calculation. For the same score, the same pre-trained model is used for both feature projection for the memorization penalty and for standard FID calculation. Inception V3 was used for public score following past literature, while the private score used NASNet (Zoph et al., 2018). We will discuss how NASNet was selected in Section 3.2.1.

### 3.2 DETERMINING FINAL RANKS

After the competition was closed to submission there is a two-week window to re-process all the submissions and remove ones violating the competition rules (e.g. by intentionally memorizing the training set) before the final private leaderboard was announced. The memorization penalty term in MiFID was efficiently configured for re-running with a change of the parameter $\tau$, allowing finalizing of results within a short time frame.

### 3.2.1 SELECTING PRE-TRAINED MODEL FOR THE PRIVATE SCORE

As it is commonly assumed that FID is generally invariant to the projection space, the pre-trained model for private score was selected to best combat cheating via training set memorization. The goal is to separate cheating and non-cheating submissions as cleanly as possible. We calculate the memorization distance for a subset of submissions projected with the chosen pre-trained model and coarsely label whether the submission intentionally memorized training samples. Coarse labeling of submissions was achieved by exploiting competition-related clues to obtain noisy labels.

There exists a threshold $\tau^*$ that best separates memorized versus non-memorized submissions via the memorization distance (see Figure 1). Here we define the memorization margin $d$ of pre-trained

model $M$ as

$$d(M) = \min_{\tau} \sum_{\forall S_g} (s(S_g, S_t) - \tau)^2$$

The pre-trained model with largest memorization margin was then selected for calculation of the private score, in this case, NASNet (Zoph et al., 2018), and the optimal corresponding memorization penalty $m_\tau$ where $\tau = \tau^*$.

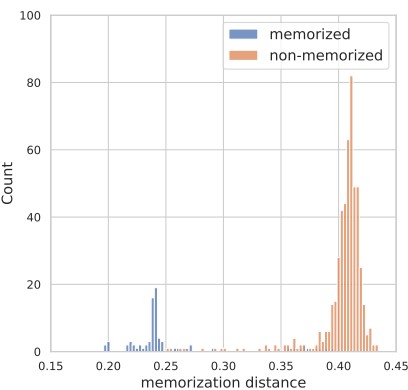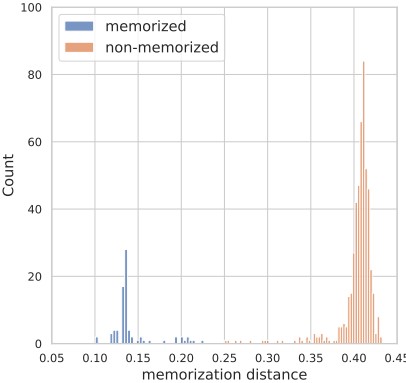

Figure 1: Histogram of memorization distance for private (left) and public (right) leaderboards (using NASNet and Inception). As shown in the figure, the two classes (legitimate models and memorizing/cheating models) are well separated.

### 3.2.2 HANDLING FALSE PENALIZATION

While MiFID was designed to handle penalization automatically, in practice we observed minor mixing of cheating and non-cheating submissions between the well-separated peaks (Figure 1). While it is well accepted that no model can be perfect, it was necessary to ensure that competition was fair. Therefore, different strategies were adopted to resolve false positives and negatives. For legitimate submissions that are falsely penalized (false positives), participants are allowed to actively submit rebuttals for the result. For cheating submissions that are dissimilar enough to the training set to dodge penalization (false negatives), the code was manually reviewed to determine if intentional memorization was present. This manual reviewing process of code submissions was labor intensive, as it required expert knowledge of generative modeling. The goal was to review enough submissions such that the top 100 teams on the leaderboard would be free of cheaters, since we reward the top 100 ranked teams. Thanks to our design of MiFID, it is possible to set the penalty threshold $\tau$ such that we were comfortable that most users ranked lower than 100 on the leaderboard who cheated with memorization were penalized by MiFID. This configuration of MiFID significantly reduced the time needed to finish the review, approximately by 5x. The results of the manual review is presented in Section 4.2.

## 4 RESULTS AND DATA RELEASE

A total of 924 teams joined the competition, producing over 11,192 submissions. Visual samples from submitted images are shown in the appendix.

### 4.1 DATA RELEASE

The complete dataset will be released with the publication of this paper to facilitate future work on benchmarking generative modeling. It includes:

- A total of 11,192 submissions, each containing 10,000 generated dog images with dimension $64 \times 64 \times 3$.

- Manual labels for the top 1000 ranked submissions of whether the code is a legitimate generative method and the type of illegitimacy involved if it is not. This was extremely labor-intensive to obtain.

- Crowd-labeled image quality: 50,000 human labeled quality and diversity of images generated from the top 100 teams (non-memorized submissions).

We will also release the code to reproduce results in the paper.

## 4.2 MEMORIZATION METHODS SUMMARY

The 1000 top submissions are manually labeled as to whether or not (and how) they cheated. As we previously predicted, the most pronounced way of cheating was training sample memorization. We observed different levels of sophistication in these methods - from very naive (submitting the training images) to highly complex (designing a GAN to memorize). The labeling results are summarized in Table 1.

Table 1: Training sample memorization methods

| METHOD | DESCRIPTION |
|---|---|
| MEMORIZATION GAN (MGAN) | MEMORIZATION GANS ARE PURPOSELY TRAINED TO MEMORIZE THE TRAINING SET WHILE MAINTAINING THE ARCHITECTURE OF A TYPICAL GAN BY MODIFYING THE UPDATE POLICY OF THE GENERATOR AND DISCRIMINATOR. THE TRAINING PROCESS IS SPLIT INTO TWO PARTS: (1) TRAIN DISCRIMINATOR ONLY WITH REAL TRAINING IMAGES (DEGENERATES TO CLASSIFIER OF TRAINING SET MEMBERSHIP), (2) TRAIN GENERATOR ONLY WITH FIXED DISCRIMINATOR. |
| SUPERVISED MAPPING (SUP) | CONSTRUCTS PAIRS OF LABELED DATA CONSISTING OF NOISE VECTORS AND TRAINING SAMPLES AND TRAIN A NEURAL NETWORK TO LEARN THE MAPPING. |
| AUTOENCODER (AE) | AUTOENCODERS ARE RELATIVELY STRAIGHTFORWARD BY TRAINING DIRECTLY ON TRAINING SAMPLES AND RECONSTRUCTING THEM. |
| AUGMENTATION (AUG) | COMBINATIONS OF TYPICAL IMAGE AUGMENTATION TECHNIQUES SUCH AS CROPPING, MORPHING, BLENDING AND ADDITIVE NOISE. THE NAIVETY OF THIS APPROACH MAKES IT THE EASIEST TO IDENTIFY AND GENERALLY CAN BE FILTERED OUT WITH THE MIFID. |

## 4.3 COMPETITION RESULTS SUMMARY

In Figure 2 (left), we observe that non-generative methods score extremely good (low) FID scores on both the public and private leaderboard. Specifically, memorization GAN achieved top tier performance and it was a highly-debated topic for a long time whether it should be allowed in the competition. Ultimately, memorization GAN was banned, but it serves as a good reminder that generative-looking models may not actually be generative. In Figure 2 (right), we observe that the range of memorization calculated by NASNet (private) spans twice as wide as Inception (public), allowing easier profiling of cheating submissions by memorization penalty. It reflects the effectiveness of our strategy selecting the model for calculating private score.

Participants generally started with basic generative models such as DCGAN (Radford et al., 2015) and moved to more complex ones as they grow familiar with the framework. Most notably BigGAN (Brock et al., 2018), SAGAN (Zhang et al., 2018) and StyleGAN (Karras et al., 2019a) achieved the most success. Interestingly, one submission using DCGAN (Radford et al., 2015) with spectral-normalization (Miyato et al., 2018) made it into top 10 in the private leaderboard, suggesting that different variations of GANs with proper tuning might all be able to achieve good FID scores (Lucic et al., 2017).

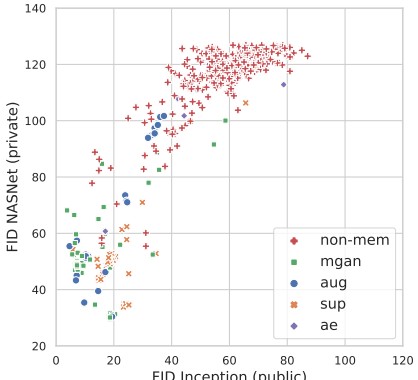 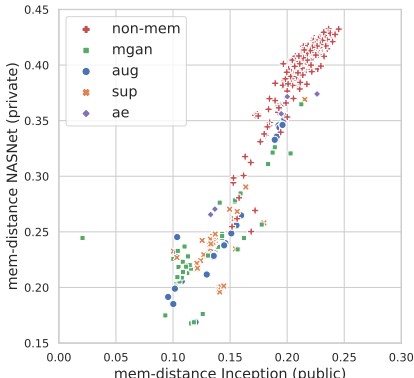

Figure 2: Distribution of FID (left) and memorization distances (right) for public vs private scores with manual labels. The better (lower) FIDs are the ones using various memorization techniques.

# 5 INSIGHTS

## 5.1 UNINTENTIONAL CHEATING: MODELS WITH BETTER FID MEMORIZE MORE

In our observation, almost all cheating submissions attempted to cheat by memorizing the training set. This is likely because it is well-known that memorization achieves a good FID score. The research community has long been aware that memorization can be an issue for the FID metric. However, there has been no formal studies on the impact of IS or FID scores affected by memorization. This can pose a serious problem when researchers continue to claim state-of-the-art results based on improvements to the FID score if there is not a systematic way to measure and address training set memorization. With disturbing findings from our study, we caution the danger of ignoring memorization in research benchmark metrics, especially with unintentional memorization of training data.

In Figure 3 (right) we plot the relationship between FID and memorization distance for all 500 non-cheating models in the public and private leaderboard, respectively. Note that these models are non-cheating, most of which popular variants of state-of-the-art generative models such as DCGAN and SAGAN recently published in top machine learning conferences. Disturbingly, the Pearson correlation between FID and memorization distance is above 0.95 for both leaderboards. High correlation does not imply that memorization solely enables good model performance evaluated by FID but it is reasonable to suspect that generation of images close to the training set can result in a high FID score.

It is important for us to take memorization more seriously, given how easy it is for memorization to occur unintentionally. The research community needs to better study and understand the limitation of current generative model benchmark metrics. When proposing new generative techniques, it is crucial to adopt rigorous inspections of model quality, especially regarding training sample memorization. Existing methods such as visualizing pairs of generated image and their nearest neighbors in the training dataset should be mandatory in benchmarks. Furthermore, other methods such as the FID and memorization distance correlation (Figure 3) for different model parameters can also be helpful to include in publications.

## 5.2 DEBUNKING FID: CHOICE OF LATENT SPACE FOR FEATURE PROJECTION IS NON-TRIVIAL

In the original paper where FID is proposed (Heusel et al., 2017), features from the coding layer of an Inception model are used as the projected latent space to obtain "vision-relevant" features. It is generally assumed that Fréchet Distance is invariant to the chosen latent space for projection as long as the space is "information-rich", which is why the arbitrary choice of the Inception model has been widely accepted. Interestingly, there has not been much study on the extent of our knowledge as to whether the assumption holds true even though a relatively large amount of new generative model architectures are being proposed (many of which rely heavily on FID for performance benchmarking).

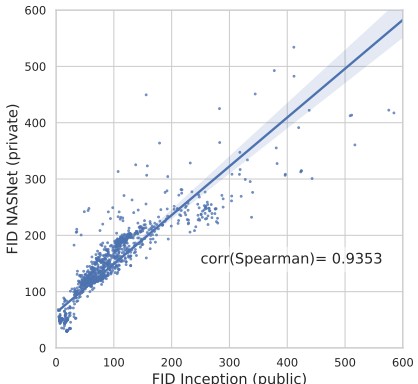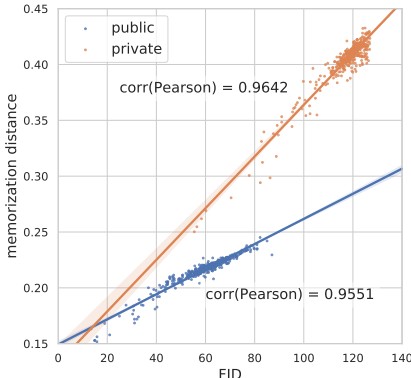

Figure 3: Public FID Inception vs private FID NASNet (left) and FID vs memorization distance distribution with non-memorized submissions (right). It shows that FID is highly correlated to memorization.

In our competition, we used different models for the public and private leaderboards in an attempt to avoid models which "overfit" to some particular feature space.

In Figure 3 (left), we examine the relationship between Fréchet Distance calculated by two different pre-trained image models that achieved close to state-of-the-art performance on ImageNet classification (specifically, Inception (Szegedy et al., 2016) and NasNet (Zoph & Le, 2016)). At first glance, a Spearman correlation of 0.93 seems to support the assumption of FID being invariant to the projection space. However, on closer inspection we noticed that the mean absolute rank difference is 124.6 between public and private leaderboards for all 1675 effective submissions. If we take out the consistency of rank contributed by intentional memorization by considering the top 500 labeled, non-memorized submissions only, the mean absolute rank difference is as large as 94.7 (18.9 %). To put it into perspective, only the top 5 places receive monetary awards and there is only 1 common member between the top 5 evaluated by FID projected with the two models.

It's common to see publications claiming state-of-art performance with less than 5% improvement compared to others. As summarized in the Introduction section of this paper, generated model evaluation, compared to other well-studied tasks such as classification, is extremely difficult. Observing that model performance measured by FID fluctuates in such great amplitude relative to the improvement of many newly proposed generation techniques, we would suggest taking progression on the FID metric with a grain of salt.

## 6 CONCLUSIONS

We summarized our design of the first ever generative modeling competition and shared insights obtained regarding FID as a generative modeling benchmark metric. By running a public generative modeling competition we observed how participants attempted to game the FID, specifically with memorization, when incentivized with monetary awards. Our proposed Memorization-Informed Fréchet Inception Distance (MiFID) effectively punished models that intentionally memorize the training set which current popular generative modeling metrics do not take into consideration.

We shared two main insights from analyzing the 11,000+ submissions. First, unintentional training sample memorization is a serious and possibly widespread issue. Careful inspection of the models and analysis on memorization should be mandatory when proposing new generative model techniques. Second, contrary to popular belief, the choice of pre-trained model latent space when calculating FID is non-trivial. The top 500 labeled, non-memorized submission mean absolute rank difference percentage between our two models is 18.9 %, suggesting that FID is rather unstable to serve as the benchmark metric for new studies to claim minor improvement over past methods.

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

## A  APPENDIX

Table 2: Public and Private leaderboards MiFID configurations

|  | PUBLIC | PRIVATE |
|---|---|---|
| MODEL | INCEPTION | NASNET |
| DATASET | IMAGENET DOGS 120 BREEDS, 20579 IMAGES | IMAGENET DOGS + PRIVATE DOGS + INTERNET DOGS |

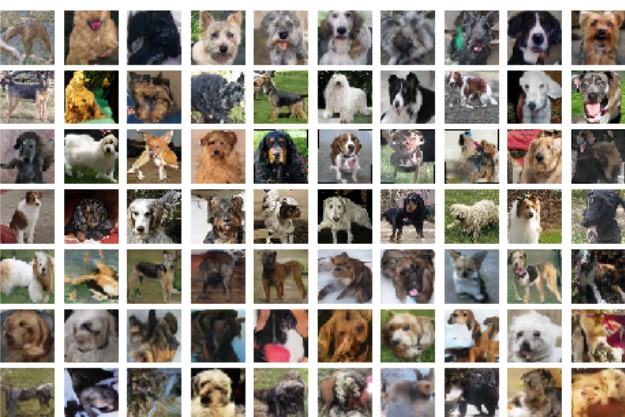

Figure 4: Submissions from ranks 1 (first row), 2, 3, 5, 10, 50, 100 (last row) on the private leaderboard. Each row is a random sample of 10 images from the same team. Visually, the quality of the generated images gets lower as the ranks get higher.

