# OpenReview forum: "A Large-scale Study on Training Sample Memorization in Generative Modeling"
_ICLR.cc/2021/Conference — Reject_

### Official Review · AnonReviewer3 · 2020-10-20
**Review of "A Large-scale Study on Training Sample Memorization in Generative Modeling"**

**Rating:** 4
**Confidence:** 4

**Review:**

The paper investigates memorization/overfitting in GANs and proposes a new metric (MiFID) to identify memorization in trained GAN models automatically. The topic of memorization in GANs is an important one and certainly one that does require more work and research.

Contributions/Novelty:
- The authors introduce a new metric to automatically evaluate memorization in GANs
- A competition is held to collect data to evaluate different kinds of memorization in GANs

Background:
The background section is quite short and only looks into the IS and FID, but not into the central topic of the paper: memorization/overfitting in GANs
- What exactly is memorization in GANs? How do you define/measure it? The concept of memorization in GANs seems to be a bit obscure in some parts of the paper (see my remarks to e.g. table 1 later on)
- What about other approaches that have looked into overfitting? E.g. [1,2]

[1] Webster, R., Rabin, J., Simon, L., & Jurie, F. (2019). Detecting overfitting of deep generative networks via latent recovery. In Proceedings of the IEEE Conference on Computer Vision and Pattern Recognition (pp. 11273-11282).

[2] Meehan, C., Chaudhuri, K. & Dasgupta, S.. (2020). A Three Sample Hypothesis Test for Evaluating Generative Models. Proceedings of the Twenty Third International Conference on Artificial Intelligence and Statistics

Competition:
The section on the competition is very sparse and I would appreciate more details:
- Why are particiapnts only allowed to train for 9h? Why not more or less? Why is the training time relevant for memorization in GANs?
- Why use a new dataset of only dog images? Why not also test with other images? Why not test with TinyImageNet or something similar?
- Why use only 20,000 images? That's a pretty small dataset (which might make memorization more likely in the first place)
- Who are the participants (academics/industry, advanced/beginner) and how were participants found/recruited?
- What was the overall goal of the competition? Identify memorization problems in GANs? Find the best model given the constraints on hardware, training time, small dataset, ...?

MiFID:
- From section 3.1: "This motivated the addition of a ”memorization-aware” metric that penalizes models producing images too similar to the training set." -> How exactly is "too similar to training set" defined?
- Why do you use the cosine similarity specifically? Why not use other metrics such as LPIPS [3] or SSIM [4]? Have you tried using other metrics?
- Technically speaking the memorization distance is not a distance since it is not symmetrical, maybe call it "divergence" instead
- For the formula/calculation of MiFID: I might misunderstand: but why use the "min" and not the "max" of the cosine distance? By using min you look for the most different sample in the training set for each generated image and use that to calculate the distances; if I'm looking for memorization wouldn't it make more sense to look for the most similar image (the memorized one) and use that for calculating the distance?

[3] Zhang, R., Isola, P., Efros, A. A., Shechtman, E., & Wang, O. (2018). The unreasonable effectiveness of deep features as a perceptual metric. In Proceedings of the IEEE conference on computer vision and pattern recognition (pp. 586-595).

[4] Wang, Z., Bovik, A. C., Sheikh, H. R., & Simoncelli, E. P. (2004). Image quality assessment: from error visibility to structural similarity. IEEE transactions on image processing, 13(4), 600-612.

Results:
- Regarding memorization methods (Table 1): why are AEs considered cheating? Yes, they "memorize" the training set, but if trained correctly (e.g. VAEs) they can also generate novel images; maybe this shows that "memorization" in this context is not well-defined: memorization (for me) means that the generative model exhibits a mode-collapse towards the training images, i.e. no matter what kind of noise input you give the model it only generates training images; this is not the case for VAEs: they will not only generate training images for any noise input you give them...
- Similarly: I also don't understand why using data augmentation during training of generative models is "cheating" and leads to overfitting; actually, there are a lot of recent works that show how data augmentation can be helpful during GAN training, especially for smaller datasets as in this case (only 20,000 images) [5,6,7,8]

[5] Karras, T., Aittala, M., Hellsten, J., Laine, S., Lehtinen, J., & Aila, T. (2020). Training generative adversarial networks with limited data. arXiv preprint arXiv:2006.06676.

[6] Tran, N. T., Tran, V. H., Nguyen, N. B., Nguyen, T. K., & Cheung, N. M. (2020). Towards Good Practices for Data Augmentation in GAN Training. arXiv preprint arXiv:2006.05338.

[7] Zhao, Z., Zhang, Z., Chen, T., Singh, S., & Zhang, H. (2020). Image Augmentations for GAN Training. arXiv preprint arXiv:2006.02595.

[8] Zhao, S., Liu, Z., Lin, J., Zhu, J. Y., & Han, S. (2020). Differentiable augmentation for data-efficient gan training. arXiv preprint arXiv:2006.10738.

Insights:
- Have you tried to use this method to also evaluate current pretrained models (BigGAN, StyleGAN, etc) for memorization roblems and visualize them?
- I don't see why the high correlation between memorizsation distance and FID is a problem; this is exactly what we would expect; I am missing experiments that clearly show that memorization is actually a problem in these models, i.e. that they only generate images from the training set

General remarks:
- I'm not convinced the results of the competition are reliable to evaluate memorization problems in GANs, especially since it is not clear who participated in the competition, why there are so many restrictions that should not directly affect memorization as such (small amount of training time and only one GPU), and the dataset for evaluation is very small (20,000 images) and only contains dogs; I would at least expect experiments on larger and different datasets and an explanation why restricting the resources is important to evaluate memorization/overfitting in GANs
- I'm also not sure I understand what exactly the authors define as "cheating"/memorization: e.g. autoencoders and data augmentation are not cheating in my perspective and also do not automatically lead to memorization; this leads back to my point that I am missing a concrete working definition on memorization in GANs in this paper
- While memorization is a serious issue for generative models I am missing a clear definition of what memorization means in this context; the authors show that there is a high correlation between the FID and their memorization distance, but this is not surprising: generative models *should* generate images close to the training set (but not "from" the training set); I feel this work lacks a clear distinction between "memorization" (only generates images from the training set) and other models (such as e.g. VAEs but also GANs) that might generate images from the training set but are also capable of generating images that do not occur in the training set (often tested by doing interpolation between two images) -> a high correlation between FID and memorization distance is therefore expected, since GANs should learn to generate images that are close to the training set -> I feel there needs to be a way to automatically detect GANs that are *not* capable of generating images that are *reasonably far* away from the training set but still realistic (if the model also generated some training images I don't think this is necessarily a big issue)

Minor:
- Why is this a table 1 a table and not a simply a bullet list/enumeration?

---

> ### Author Response · Authors · 2020-11-24
> **Detailed explanation to clarify misunderstandings [4/4]**
>
> [START OF PART 4 / 4]
>
> Finally we would like to discuss the similarities and differences between our work and the two similar papers referenced. Webster et al. consider memorization of generative models in terms of the difference in ability for the model to reconstruct images from the training and testing set, different from our idea of memorization since in our humble opinion it should be perfectly fine for generative models to generate instances similar to the training set as long as it is infrequent. Furthermore, they relied on optimizing for the nearest latent code on a highly non-linear landscape (even mentioned in their paper). Thus, the results from their proposed method (evaluating MRE distribution) are generally inconclusive since the conclusion drawn might be simply caused by not effectively finding the optimal nearest neighbor latent code.
>
> Meehan et al., on the other hand, hold a very similar view of generative modeling memorization as our work. They evaluated the frequency of models generating instances that are more similar to the training set compared to the testing set while we evaluated the frequency combined with the magnitude (by taking the mean memorization distance of generated data) between the generated data and intentionally memorized instances. They divide the generative space into uniform cells and consider memorization for each individual cell separately then aggregating the result by taking the average. We directly sampled from the latent code distribution (10k samples), evaluated the nearest neighbor cosine distance (memorization distance), and averaged the results. These are similar methods each based on aligned by slightly different definitions of memorization, which might be suitable for different applications. With our to-be-published large-scale dataset, we can compare this as well as new definitions for memorization in the future and determine which to use in what scenario.
>
> In conclusion, we elaborated on a couple of details to clarify misunderstandings and clarified the definition of memorization in generative modeling and why our memorization distance is capable of capturing generative model memorization. We hope that the additional context can allow better understanding of our study.
>
> Looking forward to further discussions. Many thanks!
>
> [END OF PART 4 / 4]

---

> ### Author Response · Authors · 2020-11-24
> **Detailed explanation to clarify misunderstandings [3/4]**
>
> [START OF PART 3 / 4]
>
> ### Competition
> > Why are participants only allowed to train for 9h? Why not more or less? Why is the training time relevant for memorization in GANs?
>
> Response: To the extent of our knowledge, there is no direct connection between training time and generative model memorization. The time limit is merely a technical constraint to distribute finite computation resources among competition participants to provide a fair and closed environment.
>
> > Why use a new dataset of only dog images? Why not also test with other images? Why not test with TinyImageNet or something similar? Why use only 20,000 images? That's a pretty small dataset (which might make memorization more likely in the first place)
>
> Response: Our goal is to base the competition on a whole new dataset to prevent exploiting prior knowledge (ex. hyperparameter selection) and prevent offline model tuning, which naturally crosses popular datasets such as variants of CIFAR10 or ImageNet off the list. To guarantee fairness, the private testing dataset should not be available anywhere on the internet in case participants scrape openly available stock images to tune their models offline. Our solution is to curate photos of our colleagues’ dogs to form the private testing dataset. 20k images is actually not that small of a dataset considering all of them are from the same class compared to other popular datasets such as CIFAR10 with 6k images per class or TinyImageNet with 600 images per class.
>
> > Who are the participants (academics/industry, advanced/beginner) and how were participants found/recruited?
>
> Response: Our competition is held publicly on a well-known machine learning competition platform where participants range from novices to experts. It is completely free to sign up as a member and participate in all sorts of competitions. We provided monetary rewards for the top 10 places as well as virtual credits for the top 100 places to incentivize their participation.
>
> > What was the overall goal of the competition? Identify memorization problems in GANs? Find the best model given the constraints on hardware, training time, small dataset, ...?
>
> Response: The goal of the competition is to (1) evaluate the efficacy of MiFID in identifying intentional memorized submissions such that more generative competitions could be held in the future to perform large-scale researches on different topics, (2) identify the relationship between good performance in FID and memorization distance and provide constructive feedback to the research community, and (3) release the collected generative models as an open dataset to facilitate research on searching for better evaluation metrics or other fundamental topics.
>
> [END OF PART 3 / 4]

---

> ### Author Response · Authors · 2020-11-24
> **Detailed explanation to clarify misunderstandings [2/4]**
>
> [START OF PART 2 / 4]
>
> We will proceed to reply to the rest of the comments point by point:
>
> ### MiFID
> > How exactly is "too similar to the training set" defined?
>
> Response: As the optimal threshold that perfectly separates the intentional memorized and legitimate submissions does not exist, we aim to follow a good enough heuristic to help us reject most intentionally memorized models and significantly reduce manual review. How we chose the threshold is elaborated in section 3,2,1 and submissions with memorization distance less than the threshold is considered “too similar” to the training set.
>
> > Why do you use the cosine similarity specifically? Why not use other metrics such as LPIPS [3] or SSIM [4]? Have you tried using other metrics?
>
> Response: We experimented with cosine distance and different euclidean distances (all first projected onto a deep representation via a pretrained image classification model) and discovered no significant difference in their ability to discriminate between intentional memorization and legitimate submissions. As mentioned in section 3.1.1, cosine similarity is relatively computationally efficient which is beneficial in a competition setting. SSIM generally is considered relatively outdated as projection onto deep latent space is more representative of the semantic (as SSIM still operates in pixel space) and it is also not suitable when large geometric distortion exists between the compared images [1]. Interestingly LPIPS is equivalent to the cosine distance when applying the unit weight vector and since we have no presumptions regarding the generated images other than it should be semantically different from the training data points, applying the unit weight vector should be relatively unbiased. Furthermore, LPIPS averages the calculated distance over multiple layers but we are concerned about lower level similarity (distances produced by shallower layers) dominating the overall distance (ex. when images are cropped) so selecting the last layer ensures capturing the high level semantics. This also allows the sharing of latent representation with FID such that similarity evaluations of both distances (cosine similarity for memorization distance and wasserstein distance for FID) are consistent.
>
> [1] M. P. Sampat, Z. Wang, S. Gupta, A. C. Bovik, and M. K. Markey. Complex wavelet structural similarity: A new im- age similarity index. TIP, 2009. 2, 7, 14
>
> > Technically speaking the memorization distance is not a distance since it is not symmetrical, maybe call it "divergence" instead
>
> Response: Totally agreed.
>
> > For the formula/calculation of MiFID: I might misunderstand: but why use the "min" and not the "max" of the cosine distance? By using min you look for the most different sample in the training set for each generated image and use that to calculate the distances; if I'm looking for memorization wouldn't it make more sense to look for the most similar image (the memorized one) and use that for calculating the distance?
>
> Response: That is indeed a typo in the paper and you are absolutely correct. Thanks for paying such attention to details, going over all the nooks and crannies!
>
> [END OF PART 2 / 4]

---

> ### Author Response · Authors · 2020-11-24
> **Detailed explanation to clarify misunderstandings [1/4]**
>
> [START OF PART 1 / 4]
>
> We appreciated your extremely detailed review. That being said, there are a couple of misunderstandings that we wish to clarify and hopefully in the end project a more complete picture with the gaps filled.
>
> First we would like to clarify the most severe misunderstanding in the review, the competition results. Some variants of autoencoders are perfectly legitimate generative modeling methods such as VAE when the generation process involves randomly sampling a latent code and decoding it back to the original space. The ones that we consider cheating directly passed training instances into the encoder, then passed the compressed latent code through the decoder, and submitted the reconstructed instances as their generated results. On the other hand, of course data augmentation techniques are perfectly common for training any machine learning models and are absolutely allowed in the competition. What’s not allowed is directly submitting the augmented training instances as their generated results. There are all sorts of tricks up the competition participants’ sleeves but fortunately we are able to identify the intentional memorization submissions efficiently with the assistance of MiFID.
>
> We would then like to clarify the definition of memorization in our study and why the memorization distance/divergence is capable of capturing it as this is a frequent concern raised throughout the review comment. Memorization is different from overfitting in the sense that models that overfit usually perform poorly on the benchmark metric while models that memorize exhibit exceptional benchmark metric performance (on holdout data). A good benchmark metric is capable of identifying overfitting while an inferior one suffers from the inability to identify memorization.
>
> In general, we completely agree that a generative modeling metric should correlate with the memorization distance. As you also stated, good quality generations should be “from” the training data distribution, which should naturally locate closer to the training data points. However, the correlation between the metric and memorization distance should decrease as a model learns the training distribution since it is not ideal to reward models that produce instances closer to training points and punish models that albeit still generate instances from the training distribution, lies further away from the training data points. In this case, borrowing your terminology, the metric is oblivious to memorization when it is unable to discriminate instances generated “close to” the training distribution or “from” the training set. We concluded that the choice of FID as the benchmark metric allows models to gain unfair advantage by memorizing the training set since, as shown in our study, high correlation exists between FID and memorization distance even for the top 10% best performing submissions.
>
> Although we are definitely not the first to point out FID’s memorization issue, we are the first to provide robust empirical evidence by observing the phenomenon on a giant pool (10k+ submissions) of diverse generative models (including the most popular GAN variations such as BigGAN, StyleGAN, and ProGAN) collected from the competition. The memorization distance defined as average 1 - nearest neighbor cosine similarity captures the common practice of visualizing nearest neighbors in the training data but in a more efficient manner for a competition setting. Fig 1 in the paper shows a clear gap between intentionally memorized and normal submissions, further indicating that memorization distance does indeed capture memorization.
>
> [END OF PART 1 / 4]

---

### Official Review · AnonReviewer2 · 2020-10-28

**Rating:** 3
**Confidence:** 4

**Review:**

This paper studied the memorization issue of generative modeling. It proposed a benchmark for a public generative modeling competition and observed how participants attempted to game the FID.

It seems the paper only targeted FID. Actually in GAN evaluation, FID is not the only metric that are widely used. In addition, I doubt how the proposed competition can show the impact of memorization. Also, it is not clear how MiFID really evaulates the generilization ability. In addition, the title mentioning generative modeling is overclaimed (only GANs).

The writting of the paper is not good. The introducation spent too many paragraphes on describing the background of GANs but failed to clarify the motivation/intuiation clearly. It also used several offencive words such as "cheating".

Overall, I think this work has limit impact to the community and not provided deep insights to the audience. Based on these facts, I make my rating.

---

> ### Author Response · Authors · 2020-11-24
> **Official Rebuttal of Paper Review**
>
> We appreciated your concise remarks and would like to clarify a couple of misunderstandings.
>
> > It seems the paper only targeted FID. Actually in GAN evaluation, FID is not the only metric that is widely used.
>
> Response: We are well aware of the many metrics for generative modeling evaluation (as covered in the second paragraph of page 2) with IS and FID being the most popular in recent studies. FID is widely considered as an improvement over IS due to its better robustness to noise [1] and sensitivity to mode intra-class mode collapse [2]. Furthermore, there are already studies investigating the flaws of IS with not reporting memorization being one of them [3]. The above reasons lead us to narrow our focus on studying FID.
>
> [1] Heusel, Martin, et al. "Gans trained by a two time-scale update rule converge to a local nash equilibrium." Advances in neural information processing systems. 2017.
>
> [2] Borji, Ali. "Pros and cons of gan evaluation measures." Computer Vision and Image Understanding 179 (2019): 41-65.
>
> [3] Barratt, Shane, and Rishi Sharma. "A note on the inception score." arXiv preprint arXiv:1801.01973 (2018).
>
> >  I doubt how the proposed competition can show the impact of memorization.
>
> Response: Through the competition, we collected a great variety of diverse generative models, each tuned to its best since competition participants are incentivized to do so. The diversity and quality of the models available far surpasses that of any individual study which enables objective observations and conclusions to be drawn. In our study, we observe from the competition submissions that FID is incapable of separating models with very good generation quality and training set memorization.
>
> >  Also, it is not clear how MiFID really evaluates the generalization ability.
>
> Response: We didn’t claim that MiFID is capable of evaluating generalization. It is an effective engineering design that helps us successfully hold the first ever public, large-scale generative modeling competition. We, on the other hand, relied on the cosine memorization distance to capture the degree of memorization (or overfitting/generalization). Generating instances closer to training data points suggests that the model suffers from more severe memorization.
>
> > In addition, the title mentioning generative modeling is overclaimed (only GANs).
>
> Response: In our humble opinion, the title is, in fact, NOT an overclaim since the competition does not limit submissions to be GANs. There are other modeling methods such as different variations of autoencoders as well. A great majority of the submissions are GANs probably because it is proven to be a strong baseline. In general, our discussion and conclusions regarding memorization can be applied to any generative method evaluated by FID.
>
> > The introduction spent too many paragraphs on describing the background of GANs but failed to clarify the motivation/intuition clearly.
>
> Response: Thank you for pointing it out. Our intention is to lay down the background to  motivate the necessity of benchmarking generative methods in the real world to examine flaws of popular generative modeling metrics. We will consider making the background more concise and highlight the motivations more.
>
> > It also used several offensive words such as "cheating".
>
> Response: We use “cheating” because the competition rule actually stated that intentional memorization, such as supervised learning noise vectors that map to the training set, is prohibited. We are open to adopting a different wording but we haven’t found a better one yet.
>
> To recapitulate, our main contributions and impacts are (1) held the first ever public, large-scale generative modeling competition, (2) provided quantitative empirical evidence for FID’s inability to capture memorization, (3) identified the dependence of FID on the projected latent space, and (4) plan to publish a large-scale public dataset of generative models to facilitate future research on generative modeling benchmarks.
>
> Looking forward to further discussions. Many thanks!

---

### Official Review · AnonReviewer4 · 2020-10-31
**Promising study assessing how memorization can help game metrics for generative models**

**Rating:** 5
**Confidence:** 4

**Review:**

**Summary**
Motivated by the observation that prevalent metrics (Inception Score, Frechet Inception Distance) used to assess the quality of samples obtained from generative models are gameable (due to either the metric not correlating well with visually assessed sample quality or the metric being susceptible to training sample memorization), the authors conduct a large scale “controlled” study to assess the gameability of said metrics. The authors conducted a competition and subsequently analyzed how approaches tend to cheat so as to obtain higher FID scores. Furthermore, to assess the extent of memorization w.r.t. the FID score, the authors propose a new metric — Memorization-Informed Frechet Inception Distance (MiFID) — which takes into account sample memorization w.r.t. a reference set. The authors conclude on a few notable observations — (1) unintentional memorization in generative models is a serious and prevalent issue; (2) the choice of latent space used to compute FID based scores can make a significant difference.

**Strengths**
- The paper is generally well-written and easy to follow for the most part. The authors do a good job of walking the reader through the design of the competition, the proposed metric and design choices adopted to ensure fair characterization of sample-memorization and cheating strategies.
- The choice of memorization assessment defined and adopted for the FID score in section 3.1 is intuitive and well-motivated. I particularly appreciated how the authors identified the fact that assigning a memorization penalty is itself subjective to the choice of feature space, reference set and the memorization threshold. Furthermore, the constraints assigned on the challenge submissions seem reasonable to ensure fair comparisons across multiple submissions. I would also like to highlight that in addition to automating “memorization” detection to the extent possible, the authors went the extra mile to manually review code-submissions to further identify false negatives — all leading to a concise characterization of the strategies adopted to cheat or memorize.
- I think this large-scale study (although operating under several constraints due to it’s controlled nature) is going to be quite beneficial to the community. Careful inspection of the submissions reveals several insights (and caveats) associated with current metrics / results for generative models that can perhaps motivate the community to take measures to establish more reliable benchmarks for generative modeling.

**Weaknesses**
I will mostly highlight weaknesses and other points that likely have clarity issues associated in the current draft.
- While the choice of constraints imposed on the challenge submissions for the study here seem reasonable, I’m concerned the study only highlights issues associated with approaches that can likely be executed / implemented under these constraints (pointing to limited computation time, isolated containerization and restricted access to pre-trained models or additional data). While this is a good starting point, it’s unclear how well insights from the study may generalize to approaches that operate outside of these constraints. Do the authors have any thoughts on this?
- In addition to other factors, the choice of the memorization margin (as described in section 3.2.1) seems to depend not only on the reference set but also on the “sets” of generated images. This likely affects all the following stages of re-calibrating the score after identifying false-positives and negatives. Can the authors comment on how dependent is this on the number and kind of submissions? This will likely inform the extent to which the analysis setup (including the metric) can be adopted / extended to other kinds of images, datasets, etc (and maybe even more relaxed constraints).
- Minor comments — (1) The paper would benefit from making the distinction between intended and unintended memorization clear early on in the introductory section; (2) I’m curious about how the constraints posed in the section 3 on challenge were verified.

---

> ### Author Response · Authors · 2020-11-24
> **Clarifying conclusions drawn from the competition should generalize to non-competition settings**
>
> We really appreciate our sharing of the same view on the importance of studying memorization in generative modeling.
>
> Regarding the first issue, we believe that the conclusions can indeed generalize as the restrictions are not related to memorization. Specifically, longer training time is usually associated with more overfitting thus would likely exacerbate the memorization issue. Using pre-trained models is more common for other tasks (ex. classification) and less so (if any) for the generative case while additional data is generally not allowed in generative settings anyway.
>
> Regarding the second issue, we agree that the margin is in fact generated-set dependent and is difficult to theoretically prove its applicability to other datasets. That being said, we consistently observe a clear divide between memorized and non-memorized instances on memorization distance for different pretrained model projection (such as InceptionV3 and NasNet in the paper) as well as for different memorizing methods (see fig 2 right). In terms of number of samples, we found that having 100 labeled submissions worked decently well.
>
> We agree that clarifying the definition of intentional memorization is beneficial to understanding the work but are not sure what constraints specifically are you referring to in section 3?
>
> Looking forward to further discussions. Many thanks!

---

### Decision · Program_Chairs · 2021-01-07
**Final Decision**

**Decision:**

Reject

**Comment:**

The paper proposes a competition on generative models on a new dataset to study memorization in generative models and propose a  new metric Memorization-Informed Frechet Inception Distance (MiFID).

While this is an important topic, reviewers raised multiple issues and concerns regarding 1)  the metric definition (that it needs to be max and not min, this was acknowledged in the rebuttal but not updated in the paper) , 2)  how this competition is ran in terms of the definition of "cheating",  that the setup is not controlled and only constraining the time of training 3)  the notion of MiFID is depending on the sets of samples considered and the feature extractor used.

Some other reviewers raised concerns that the paper is only concerned by FID and not other metrics , and that it was only verified on GANs.  We hope the authors will address those concerns and submit the paper to an upcoming venue.